# Effects of Physical Exercise Program in Adults with Intellectual and Developmental Disabilities—A Study Protocol

**DOI:** 10.3390/jcm11247485

**Published:** 2022-12-16

**Authors:** José Pedro Ferreira, Rui Matos, Maria João Campos, Diogo Monteiro, Raul Antunes, Miguel Jacinto

**Affiliations:** 1Faculty of Sport Sciences and Physical Education, University of Coimbra, 3040-248 Coimbra, Portugal; 2Research Center for Sport and Physical Activity (CIDAF), 3040-248 Coimbra, Portugal; 3ESECS–Polytechnic of Leiria, 2411-901 Leiria, Portugal; 4Life Quality Research Centre (CIEQV), 2040-413 Leiria, Portugal; 5Research Center in Sport Sciences, Health Sciences and Human Development (CIDESD), 5001-801 Vila Real, Portugal; 6Center for Innovative Care and Health Technology (ciTechCare), Polytechnic of Leiria, 2411-901 Leiria, Portugal

**Keywords:** physical fitness, cognitive capacity, intellectual disability, physical exercise, quality of life

## Abstract

We developed a physical exercise (PE) program for people with Intellectual and Developmental Disabilities (IDD), aiming to determine the effects on physical fitness, health, cognitive ability, and quality of life (QoL). Using experimental methodology, this intervention study recruited 21 adults (18 to 65 years old), institutionalized and with no other associated pathology, who will be allocated to one of the different groups: (i) gym/indoor intervention group (using weight machines), (ii) outdoor intervention group (using low-cost materials), or (iii) control group (without specific intervention, who continue with their normal daily activities). Both intervention groups will engage in 45 min of training per session, twice a week, for 24 weeks. Assessments will be conducted at baseline (initial assessment), 3 months (mid-term assessment), and 6 months (final assessment). Variables assessed include anthropometrics, body composition, functional capacity, muscle strength, general health, cognitive ability, and QoL. The results of this study will assist in the development of more effective strategies, recommendations, and interventions to ensure better and greater adherence to PE by institutionalized individuals with IDD, namely, recommendations for assessment, prescription, and implementation of PE for this population. Additionally, we intend to make available two PE programs, if they are adapted and promote positive effects.

## 1. Introduction

Individuals with Intellectual and Developmental Disabilities (IDD) (new designation according to *ICD-11 International Classification of Diseases 11th Revision*) [1] are characterized by a deficit of intellectual and adaptive functioning in the conceptual, social, and practical domains. This individual can be identified with mild, moderate, severe, and profound degrees and are a type of disability that develops before age 22 [2,3].

In this population, several studies show improvements in physical fitness through the implementation of physical exercise (PE) programs, increasing the success in performing activities of daily living [4,5]. Performed in a regular way, PE is also associated with brain development, particularly the frontal and temporal region, due to the cortical activity it stimulates [6], affecting brain plasticity and improving cognitive ability [7,8]. In individuals with IDD, a simple 20 min treadmill walking training provides significant improvements in choice response time and processing speed, attention, and inhibition [9,10]. In the same sense, the improvements found are not only in terms of physical fitness and cognitive ability variables, but also in quality of life (QoL) [11]. The QoL of this population is a construct consisting of a set of factors that determine their well-being, or their perception of their social position, including individual needs, experiences, and sociocultural values and preferences [12,13]. In the Pérez-Cruzado and Cuesta-Vargas [14] study, an 8-week PE intervention program promoted physical fitness and QoL in 40 individuals. In addition, a previous study with a sample of 529 individuals with IDD, despite being an intervention with physical activity, identifies benefits in each of the domains of the aforementioned model [15].

Despite the importance of PA and/or PE benefits, this population has high rates of sedentarism and low levels of physical activity engagement [16,17], not meeting the recommendations [18,19], which puts them at greater risk of developing chronic diseases [20] and consequent premature death [21,22]. The risks of physical inactivity in people with disabilities increase when compared to the general population: higher prevalence of hypertension (25 to 41% compared to 29% in the general population); obesity (30.8 to 36.6% compared to 18.2 to 18.5%); and metabolic syndrome (21 to 29.3% compared to 13.5% to 17.9%) [23,24]. In addition, as the degree of disability increases, physical activity levels decrease more significantly [25], affecting their physical fitness, capacity to perform activities of daily living, and independence [26]. To promote their life conditions, people with IDD should be integrated into PA and PE practice, according to recent recommendations [18,19].

One of the possible reasons for these low levels of practice is the existence of barriers that hinder their practice. Among them, we can find factors such as financial cost and lack of adapted PE [27,28]. To reduce the impact of these barriers, it is important to develop PEs. Besides being adapted to the individuals, they must be accessible to any type of context and financial availability, both for the individuals and the support institutions. In addition, the development of these PE may contribute to the reduction in the risk of the onset of metabolic and cardiovascular diseases, which decreases healthcare costs and promotes physical fitness, functionality, and QoL. Therefore, the main objective of the present study is to evaluate and compare the effectiveness of two different 24-week PE programs on levels of physical fitness, general health, dementia/cognitive decline, and QoL in individuals with IDD.

Therefore, the following hypotheses will be operationalized: (i) individual with IDD increased physical fitness after 12 and/or 24 weeks of PE; (ii) individual with IDD increased general health after 12 and/or 24 weeks of PE; (iii) individual with IDD increased dementia/cognitive decline after 12 and/or 24 weeks of PE; (iv) individual with IDD increased QoL decline after 12 and/or 24 weeks of PE; (v) there was a difference between groups on levels of physical fitness after 12 and/or 24 weeks of PE; (vi) there was a difference between groups on levels of general health after 12 and/or 24 weeks of PE; (vii) there was a difference between groups on levels of dementia/cognitive decline after 12 and/or 24 weeks of PE; and (viii) there was a difference between groups on levels of QoL after 12 and/or 24 weeks of PE.

## 2. Materials and Methods

### 2.1. Study Design

The present study protocol describes a non-randomized experimental study, consisting of three groups (1:1 allocation), to assess and compare the effectiveness of two different combined PE (strength + aerobic capacity) for individuals with IDD.

Participants will be allocated to one of the three groups: (i) indoor training group (IG) with sessions carried out in a gym, using weight machines; (ii) outdoor training group (OG) with sessions using low-cost materials; and (iii) control group (CG) with participants continuing to do their normal activities (participation in one of the exercise interventions will not be allowed), based on their interests and availability to be involved with PE programs.

Participants from both intervention groups will participate twice a week, in a 24-week supervised PE program, with sessions lasting 45 min per session. All outcome measures will be collected at three different time moments, at baseline (time 0, baseline assessment, or week 0), at mid-term (time 1, intermidiate assessment, week 12), and at the end of the intervention program time 2, final assessment, week 24). The measurements will be performed after 48 h, after the last workout in both groups, and always in the morning. Figure 1 shows the participant flow for this trial.

### 2.2. Participants

Adult volunteers, institutionalized in a support institution, located in Leiria, Portugal, recruited by the non-probabilistic convenience method will participate in the program.

In the first step, an individual explanation will be held about the procedures and objectives of the study, as well as the potential benefits, risks, and the time needed for the development of the project.

In a second moment, the participants/family members/tutors will sign a free consent form. Inclusion criteria will be defined: (1) adults with IDD, diagnosed with mild, moderate, or severe IDD; (2) age over 18 years) success in performing movements such as pulling/pushing; (3) capacity to perform the assessments. Exclusion criteria will be defined: (1) individuals who cannot commit themselves for 6 months; (2) individuals with other associated pathologies; (3) contraindications to PE (e.g., high blood pressure); (4) inability to walk unaided; (5) profound IDD; (6) inability to communicate; (7) non-delivery of signed informed consent.

Due to the characteristics of this special population and logistical constraints intrinsic to the development of intervention studies, the sample will consist of the first 21 individuals who agree to participate in the program, aged between 18 and 65. After that, participants will be allocated to one of the three groups. A power analysis (calculated using *G*Power*, version 3.1.9.7 [29] showed that a sample of at least 15 was required to detect a medium effect size (*ES*) of 0.5 (*α* = 0.05, 1 − *β* = 0.95) using a repeated-measures analysis of variance (*ANOVA*), in agreement with some previous studies [30,31]. The effect size of 0.5 was chosen given that this value was verified from studies investigated the effects of exercise on the variables of interest in our study [32,33,34].

### 2.3. Informed Consent

A comprehensive explanation of the study (material and methods inclusive) by the research leader and the host institution will be conducted to allow participants/family members/tutors to be fully informed. The subject group will be given adequate time to decide on their participation; however, the first 21 individuals who agree to participate in the study will be part of the sample. To do this, participants/family members/tutors must sign and deliver the informed consent form.

### 2.4. Protocols

The development of 2 training programs, one Indoor (Table 1) and one Outdoor (Table 2), occurred taking into account one of the barriers identified by individuals with ID and their caregivers and/or family members/tutors to physical inactivity: lack of adapted PE and the financial cost of the practice [27].

The indoor training program should be performed in the context of a gym/health club/association, using the available materials. The outdoor training program should be performed with low-cost materials, allowing its application in any economic context. The use of some low-cost equipment is suggested, such as elastic bands and shin guards of different weights. However, if this is not possible, the outdoor PE can be performed with bodyweight only or with water/sand bottles. These programs were as close as possible to the recommendations suggested by [18], as well as to the scientific evidence [4].

Strength training was combined with aerobic training in the same session, performed twice a week, for 45 min, always at the same time, namely, in the morning. Each training session was divided into 4 phases (warm-up, cardio, resistance, and flexibility training). Considering the initial assessment and physical fitness of each participant, the intensity is progressive throughout the program, and the stimulus and load will be continuously adjusted in order to promote the consequent adaptation processes in the body, taking into account the principle of progressive overload. Although a set of exercises are suggested, the planning is not fixed and may be adjusted to the characteristics of individuals. First, the required movement will be practiced without any kind of material. All participants will have familiarization sessions with the training materials, prescribed movements, and with the space and with all the routines that an exercise session involves. Occasionally, if individuals mention/evidence any kind of mental/psychological or physical problem, the training intensity is adjusted to allow participation in the exercise session.

The programs will be supervised by two PE technicians, trained in a standardized manner, with recovery periods of at least 48 h. The control of training intensity will be carried out using heart rate monitors (Polar M400, Kempele, Finland), through the formula used to calculate the Maximum Heart Rate [35] and also the equation for individuals with Down Syndrome [18] to calculate the target heart rates. If any subject exceeds the target intensity, they will be instructed to decrease or even rest.

#### 2.4.1. Indoor Training Program

The indoor PE program was carried out in a gym with weight machines. Seven heart rate monitors and seven gym towels will be required. The PE program was divided into four parts. Part I: playful game or shuttle run (5 to 7 min). Part II: aerobic training; Part III: strength training; Part IV: 4 static stretches/cool down. Table 1 shows the training program in detail.

#### 2.4.2. Outdoor Training Program

The program will be conducted in an outdoor space with elements of nature, where resistance elastic bands and shin guards for the ankles will be needed. The outdoor PE program was carried out in a natural environment near the institution. Natural environments, which, for this experimental study, are defined as “any outdoor spaces with elements of nature, from pure or semi-natural areas to urban green or blue spaces, including green infrastructure” [36]. The PE program was divided into four parts. Part I: playful or shuttle run; Part II: aerobic training; Part III: strength training; Part IV: 4 static stretches/cool down. Progression of exercises with changing the resistance of the TheraBand’s and shin guards. Table 2 shows the Outdoor PE program in detail.

#### 2.4.3. Control Group

Subjects in the control group will be encouraged to keep their usual lifestyle and their attendance at exercise sessions corresponding to the two exercise programs will not be allowed.

### 2.5. Assignment of Intervention and Blindness

After obtaining informed consent and finishing the initial assessments, the volunteers will be distributed to the three groups, according to their intention to participate in the study. Due to the nature of the intervention, after the initial assessment, it will not be possible to randomize the sample by groups, nor will it be possible to “blind” the participants and lead investigator belonging to the group. Upon delivery of the informed consent, a code will be assigned to each participant, ensuring the anonymity of the subjects. The researchers responsible for the evaluations will have no knowledge of which group each subject belong to, except for the principal researcher. To minimize differences in procedures, the same team will perform the assessments at different moments.

## 3. Outcomes

The assessments will be carried out in the laboratory of the Faculty of Sport Sciences and Physical Education—University of Coimbra. The space is ample and isolated, the temperature controlled, and each step of the assessment should be organized to provide maximum comfort and privacy to each participant. The research team will provide information on the procedures and aims. The researchers will answer any questions that may arise. All assessments will be performed in a controlled environment, during the morning period, with only the ingestion of breakfast due to the characteristics of the participants and the taking of medications.

## 4. Instruments/Procedures

### 4.1. Anthropometry

For the measurement of body mass and height, a scale with a portable stadiometer model seca (model 870, Hamburg, Germany) will be used. The participant will stand barefoot on top of the stadiometer platform, leaning against the pole of the device, looking forward, and with arms along the body. Subsequently, the Body Mass Index of the formula, weight (kg)/height (m^2^) will be calculated and the waist circumference will be measured, measured halfway between the iliac crest and tenth rib, directly on the skin, using a flexible tape measure; the methods being viable, reliable, and accurate for the population [37].

### 4.2. Body Composition

For the evaluation of body composition, bio-impedance equipment (InBody770) will be used, being a viable, reliable, and non-invasive method [38]. The participant must climb to the platform of the device with bare feet, to contact the four electrodes of the feet, measure the weight, and hold the bar with the four electrodes of the hands.

### 4.3. Neuromuscular Capacity

Lower limb strength will be assessed using an isokinetic dynamometer (BIODEX Multijoint System 3 Pro, Shirley, NY, USA), which is reliable for the target population [39], through leg flexion and extension, using maximal concentric contractions. Equipment calibration was performed before the evaluation session according to the manufacturer’s instructions (Biodex Medical Systems, Inc., 2000, Shirley, NY, USA). Participants will sit in the equipment chair (chair inclined backward at 85° of hip flexion), according to the manual’s recommendations, and stabilize using crossed belts close to the chest, hip, and thigh of the member to be evaluated, in order to avoid compensation. The axis of rotation of the dynamometer was aligned with the external femoral condyle of the knee. The fixing strip of the pad was adjusted 2 cm above the upper edge of the fibular malleolus. The global range of motion was defined as between 85 and 90°. The individual gravity calibration was corrected before each test in the position of 30 degrees of knee flexion [40]. To familiarize the participants, before starting the test, 3 repetitions were performed for each velocity and action [41]. During the test, participants were instructed to keep their arms crossed with their hands on the opposite shoulder. The computer screen connected to the dynamometer provided consistent visual feedback in real time [42]. The warm-up protocol consists of 5 min of walking at a comfortable intensity. Concentric and eccentric reciprocal muscle actions will be tested considering 3 repetitions for each movement 60°.s^−1^ (1.05 rad/s) and 120°.s^−1^ (3.14 rad/s). A 60 s interval was established between the 3 rep familiarization and the test, as well as between angular velocities [43]. Values will be obtained such as peak torque, Peak TQ/BW, Maximum Repetition Total Work, Coeff of Var., Average Power, Total Work, Acceleration Time, Deceleration Time, ROM, Average Peak TQ, and Agonist/antagonist ratio.

To measure upper limb strength, a handgrip test will be used, using a manual dynamometer, the reliability and validity of which have been confirmed by Cabeza-Ruiz et al. [44] and Oppewal and Hilgenkamp [45] and the procedures recommended by the Brockport Fitness Test Manual will be used [46]. The “3 kg medicine ball throw test” will also be applied [47], valid and reliable for people with IDD [37], in order to assess the muscular power of the upper limbs. The participant will be seated in a chair with the ball close to the chest. At the start signal, throw the ball, and raise the chest pass from as far as possible.

The test of 3 maximum repetitions, for the prescribed actions/movements, will also be applied to the participants of the indoor PE program. To perform the test of 3 maximum repetitions for the participants of the IG, a small warm-up is performed, and a load is placed on the equipment, depending on the experience of the PE technician who supervises the test, so that the individual cannot perform more than 3 repetitions. If you exceed 3 repetitions, the load is gradually increased (2–5 kg) and a rest period is given between the new set (3–5 min).

### 4.4. Functional Capacity

Fullerton battery of functional tests [48] will be used to assess physical fitness, namely, the tests: “sit to stand” for 30 s, evaluating the strength and resistance of the lower limbs, a viable and reliable test for people with IDD [37,49]. The purpose of the test is to assess the strength and resistance of the lower limbs (number of executions in 30 s without using the upper limbs). The test begins with the participant sitting in the middle of the chair, with the back straight and feet shoulder-width apart and fully supported on the floor. At the “start” signal, the participant rises to maximum extension (vertical position) and returns to the initial sitting position. The participant is encouraged to complete as many repetitions as possible within a 30 s time interval.

The “agility” test to assess physical mobility, validity, and reliability will be used as assessed by Cabeza-Ruiz et al. [44], in which the objective is to assess physical mobility, namely, speed, agility, and dynamic balance. The participant must be seated in the chair, with hands on thighs and feet flat on the floor. At the starting signal, he gets up from the chair and walks as fast as possible (without running), around a cone (located at 2.44 m) and returns to the chair. The participant must be informed that the test is evaluated by the time it takes to perform the exercise.

The “6 min walk test”, to assess aerobic resistance, is also valid and reliable for the study population [50]. The objective of the 6 min walking test is to assess aerobic resistance by covering the greatest distance in 6 min. At the starting signal, the participant is instructed to walk as quickly as possible (without running) the distance marked around the cones. If necessary, participants can stop and rest, being able to sit down and resume the course.

### 4.5. General Health Status

For the evaluation of the general state of health, blood samples will be collected by professionals accredited for this purpose, through the venipuncture technique [51]. The results will be analyzed by the certified laboratory to which the professional belongs and will be sent via email to the principal investigator of the study.

A digital sphygmomanometer Omron Digital Blood Pressure Monitor HEM-907 (Omron Healthcare Europe BV, Matsusaka, Japan) will be used to obtain hemodynamic parameters, such as resting blood pressure (systolic and diastolic) and resting heart rate, as well as an oximeter. Participants, before data collection, will remain at complete rest for five minutes, with legs uncrossed, and back and arm supported without speaking/or moving [52]. Two readings will be taken, with an interval of 1–2 min between them and the average of these readings will be recorded. If the values deviate ≥ 5 mmHg, a third measurement will be taken [52]. Measurements will be taken in the morning, with only one meal and participants will be instructed to avoid caffeine, exercise, and smoking for at least 30 min before measurements [52].

The Heart Rate Variability will also be evaluated, according to the procedures of Proietti et al. [53] and the guidelines *Task Force of the European Society of Cardiology* and the *North American Society of Pacing and Electrophysiology* [54], using a Polar ProTrainer (Kempele, Finland). Participants will place the sensor on their chest, below the pectoralis major. Afterward, the participants will be instructed to sit comfortably in a chair, with their eyes open, with a calm breath, and to avoid any movement during the data acquisition period. The test will last 10 min, in a calm, silent, and low-light environment. After the test is performed, the data will be downloaded via the Polar Flow Web Service as “.txt” files and exported for analysis using the Kubios HRV software package (Kubios HRV, Biomedical Signal Analysis Group, Department of Applied Physics, University of Kuopio, Finland) [55]. The RR intervals corresponding to the first two minutes will be discarded (stabilization period) and the data from the following five minutes will be used to calculate the heart rate variability. In the time domain, the following items will be calculated: (i) mean RR (mean of the RR intervals in ms); (ii) SDNN (standard deviation of RR intervals in ms); (iii) RMSSD (root mean square of successive RR interval differences in ms); (iv) pNN50 (percentage of successive RR intervals that differ by more than 50 ms). In the frequency domain, the following items will be calculated: (i) LF (absolute power of the low-frequency band, 0.04–0.15 Hz, in ms2); (ii) HF (absolute power of the high-frequency band, 0.15–0.4 Hz, in ms2; (iii) ratio of LF-to-HF power (LF/HF).

### 4.6. Quality of Life

For the evaluation of QoL in people with IDD, the *Personal Outcomes Scale* [56,57], in the Portuguese version [58] based on the model of Schalock and Verdugo [59], developed at the *Arduin Foundation and Ghent University*, will be applied by technicians with specific training for this purpose.

The Personal Outcomes Scale includes eight domains, each containing five questions, which can be answered through self-report or the report of caregivers, who may be family members or professionals, making a total of forty questions, presented with three response options, through the Likert format, as, for example, often, sometimes, or never, thus making it possible to (1) measure the results of service interventions; (2) reorient public policies to improve these results; (3) improve support management and the service financing mechanism; (4) compare to QoL of citizens; (5) monitoring the application of human and legal rights, provides crucial information so that it can be used to improve the quality of intervention and credible and sustainable practices, important to implement strategies that align at different levels of systems [58].

### 4.7. Cognitive Function

The *Mini-Mental State Examination* is a simple test, using a sheet of paper and pencil, with an easy and quick application (about 5 to 10 min) [60] and is a cognitive deficit/dementia screening test. This test is adapted to the Portuguese population [61], has been applied by some authors to the population with IDD [62], and consists of thirty items (scored with a value of 0—when the person gives an incorrect answer or simply does not answer or 1—when the person answers correctly) organized by six domains: Orientation (which assesses recent memory, attention, and the temporal–spatial orientation); Retention (assess attention and short-term or primary memory); Attention and Calculation (assesses calculation ability, attention, and immediate and working memory); Evocation (assesses recent or secondary memory); Language (assesses spontaneous speech, listening, repetition, naming, reading, and writing). The maximum test score is thirty points, with higher scores indicating better results.

To assess cognitive ability through the *Mini-Mental State Examination*, you will need a sheet of paper and pencil. The test will take about 5 to 10 min and will be conducted by a technician with specific training for this purpose, following the protocol. As with the Personal Outcomes Scale, the application of the Mini-Mental State Examination scale will be carried out in a room without isolation from noise and possible distractions, in a 1:1 aspect (one specialist for one participant).

For the evaluation of QoL in people with IDD, the *Personal Outcomes Scale* will be applied to the self and to the reference technicians/caregivers, by specialists with specific training for this purpose, following the protocol. The application of the scale will be carried out in a room without isolation from noise and possible distractions, in a 1:1 aspect (one specialist for one participant).

## 5. Procedures

Intervention study that involves several phases: (a) design of PE programs; (b) promotion of the program (information pamphlets will be distributed to the target people with IDD/guardians/tutors/caregivers about the study procedures and aims); (c) sample recruitment (participants/parents/tutors will sign an informed consent form and all project will be carried out in accordance with the Declaration of Helsinki; (d) division according to their intention to participate in the study: (i) IG/indoor context (using weight machines), (ii) OG (low cost), or (iii) CG (no specific intervention; continue with normal daily activities); (e) initial evaluation of the groups by an evaluator blinded to the group belonging; (f) intermidiate evaluation of the groups by an evaluator blinded to the group belonging; (g) final evaluation of the groups by an evaluator blinded to the group belonging; (h) data analysis: descriptive, comparison between groups and conditions, comparison before and after, correlations, among others (subjects who do not perform at least 75% of the sessions will be excluded from data processing).

## 6. Adverse Events

Despite all safety procedures adopted/provided, participants may, at some moment, experience some adverse effects, as well as be aware of some unlikely risks, during the assessment protocols or even during the interventions. When starting a PE program, people who, as a rule, adopt a sedentary lifestyle, run the risk of experiencing some type of discomfort, fatigue, or muscle pain. Given that the intensity will be gradually increased, it is expected that muscle pain will be light and brief, mitigating with the muscle’s habit of exertion. Therefore, the leader will be available for any dialogue throughout the project, to help and advise the participants.

Likewise, participants who develop any type of physical injury or other health problem during the interventions will be referred to the institutional physiotherapy office or family doctor and their continuation in the project will depend on the consideration of all related parties. All adverse physical and psychological outcomes will be recorded, described, and reported in future publications.

## 7. Participation Attendance/Adherence

Individual participation in each PE session will be scored with 0 (zero) if the participant does not attend or 1 (one) if the participant is present at the PE session and participates in it. At the end of the interventions, the participants who do not perform at least 75% of the sessions will be excluded from data processing.

## 8. Statistical Analysis

In the statistical analysis, descriptive parameters will be used (mean ± standard deviation or percentage) normality and homogeneity will be verified through the *Shapiro*–*Wilk* test and the *Levene* tests, respectively. The existence of significant differences between groups will be analyzed using the *Kruskal*–*Wallis* test and between times using *Wilcoxon* or *Friedman* test. The non-parametric statistic was defined according to the “Central Limit Theorem” [63], in the sense that it will require 30 participants in the sample to have a normal distribution. The magnitudes of the differences will be examined using the *Cohen’s d* effect size [64]. Likewise, associations across studies variables will be verified through the *Spearman* coefficient of bivariate correlations (*r* = 0.10 to 0.29—small; *r* = 0.30 to 0.49—moderate; *r* = 0.50 to 1—strong) [65]. The significance level adopted for all analyses will be *p* < 0.05. For data processing, the computer program *Statistical Package for Social Sciences* (SPSS Science, Chicago, IL, USA), version 28 will be used.

## 9. Discussion

This study will allow us to investigate and compare the effectiveness of two different 24-week exercise programs, in different environments, on the levels of the physical fitness, general health, cognitive ability, and quality of life of individuals with IDD. We intend to evaluate a set of quantitative variables at the same time, with the expectation that participants in the intervention groups will show positive changes, taking into consideration previous studies [4,32]. In the control group, no changes in the variables analyzed are expected. It is believed that the expected results can be attributed to the physical and physiological effects of the environment associated with the different exercise protocols proposed, as there is evidence that finds disparate results in PE programs performed in different contexts [66]. The results will be published at the conclusion of the study.

The pilot study that will follow is essential to understand if these two PE programs are effective tools to reduce barriers that hinder/impede their practice [27,28], and their effectiveness in promoting the variables assessed, taking into account the low levels of physical fitness and QoL of this population. Likewise, we intend to contribute with implications for the practice with new interventions with physical exercise, prescription, and effective strategies, which we believe can contribute, at all levels, to individuals with IDD.

The expected physiological benefits are based on the hypothesis that regular PE practice leads to adaptations in both the cardiovascular and musculoskeletal systems that support an overall increase in exercise capacity and performance [66]. It is considered an ideal tool in the prevention and treatment of various types of cardiometabolic diseases, a promoter of physical fitness, performance activities of daily living, and in reducing/mitigating sarcopenia and cognition decline. In addition, neuromuscular mechanisms, namely, the increased expression of neurotrophic factors (i.e., BDNF), the increase in serotonin and norepinephrine, the regulation of the HPA axis activity, and decreased systemic inflammatory signaling [67,68,69,70,71], may be associated with cognitive decline and QoL.

This study has also a multidisciplinary approach, as it is prudent to investigate the combined effects of some independent variables. In addition, we will also examine the hypothetical premise that some objective measures have strong associations with subjective perception measures.

Although there is a clear need to carry out more research on healthy lifestyle interventions for people with more severe levels of IDD (profound IDD), our current intervention is limited to participants with mild to severe IDD, so future studies should take this level of IDD into account

Finally, as with all studies, our study has some limitations, among which we highlight: (1) due to logistical constraints, the groups will not be randomized; (2) it is impossible to control activities outside the PE program, which may affect negatively or positively the variables under study; and (3) the PE program is not adapted to individuals with profound IDD.

## 10. Ethics and Dissemination

Any changes to the protocol will be agreed upon by the research team and formally reported to the Faculty of Sport Sciences and Physical Education—University of Coimbra ethics committee prior to application. Written informed consent will be obtained from participants (who will be provided a copy) after the study has been fully explained (i.e., study procedures, objectives, potential risks, and expected outcomes). Each participant will receive a single coded identification number to maintain their confidentiality, and all experimental data will be recorded using these codes.

All data evaluated will be collected and stored strictly for research purposes. The evaluation data will be stored on a laboratory computer, in a folder with a password, to which only the principal investigator will have access and will be kept for a period of 5 years after the end of the investigation. After this period, all data will be irreversibly deleted.

The results of this study will be reported and published regardless of the magnitude or direction of effect, at the intended target of 3 to 6 months after the end date of the intervention (or an earlier date if conditions permit). Communication of the results to the public will begin with study participants, their families, and to the collaborators of the institution. Communication to the scientific community will be carried out through participation in conferences/congresses. Authorship of manuscripts resulting from this research will be based on the following criteria: significant contributions to the conceptualization or design of the research project, formal analysis and/or interpretations of the data, and critical writing and/or review and editing of the manuscript.

## Figures and Tables

**Figure 1 jcm-11-07485-f001:**
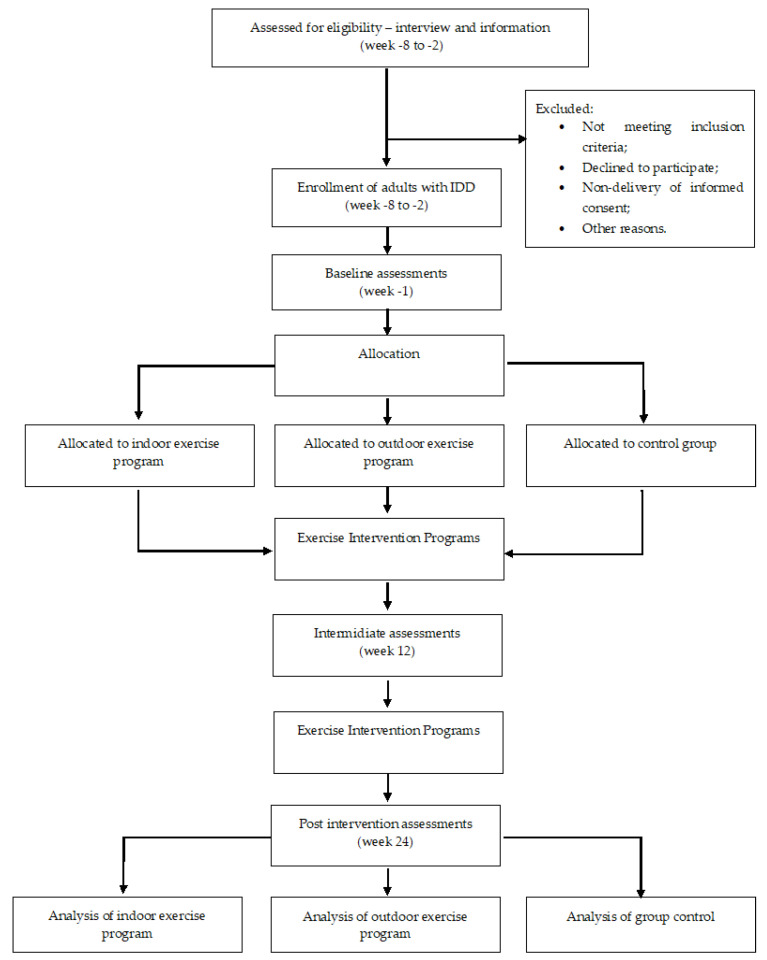
Timeline for the study design.

**Table 1 jcm-11-07485-t001:** Indoor training program.

Exercise Prescription	Weight Machines		Breathing: Continuous, Not Forced					
	Main Muscle Group/Critical Components		Duration	Speed	Intensity	Rep’s	Series	Volume
“Caterpillar game” or shuttle run	Practitioners dispersed throughout the space, an element is defined to start catching, then, who is caught, joins the colleague forming a “caterpillar” (holding hands) or “back and forth” race, which consists of executing the maximum number of exercises. routes performed at a predetermined distance and cadence.		5 to 7 min					40 min
Treadmill			10 min	Walk	40–80% of HR_max_			
Leg press	Seated; back completely against the bench; feet resting on the press at shoulder width; legs in flexion forming a 90° angle between leg and thigh; the toes in line with knee height. Perform this way the extension of the leg, without any sudden movement—Quadriceps Activation.			Max. concentric eccentric 3 s	40–80% 3RM	8–15	2–3	
Chest press	Seated; back against the bench; arms in pronation, flexed, and elbows in line with the shoulders. Carry out the extension of the same until the level of the chest. Looking forward—Activation of the Big Chest.							
Leg extension	Sitting, completely leaning against the bench; knees bent; ankles positioned on the supports. Perform the extension of the knees, until forming a parallel line between the lower limbs and the ground—Quadricipital Activation							
Lat Pull Down	Sitting in front of the machine; look ahead; slight flexion of the legs, supported on the quadriceps support; grab the handle. Perform the pull-up to the chest area—Latissimus dorsi activation							
Leg curl	Sitting and leaning against the bench; legs in extension; heels positioned on the supports. Perform knee flexion—Activation of the hamstrings							
Shoulder press	Sitting and leaning against the bench; look ahead; hands in pronation, hold the handles. Perform adduction and abduction of the shoulders, or extension and flexion of the upper limbs—Shoulder Activation							
Note: the rest interval between exercises or sets is 30 s.								
Flexibility: dynamic active method (stretching).								
Exercise	Duration (second)	Series	Recovery		Volume			
4 static stretches	30 s	1	Does not exist		2 min			
Note: stretching performed unilaterally, must last 15 s per limb (not in the case of this 1st micro cycle).								

**Table 2 jcm-11-07485-t002:** Outdoor training program.

Exercise Prescription	Resistance Elastic and 1–2 kg Shin Guards		Breathing: Continuous, Not Forced					
	Main Muscle Group/Critical Components		Duration	Speed	Intensity	Rep’s	Series	Volume
“Caterpillar game” or shuttle run	Practitioners dispersed throughout the space, an element is defined to start catching, then, who is caught, joins the colleague forming a “caterpillar” (holding hands) or “back and forth” race, which consists of executing the maximum number of exercises routes performed at a predetermined distance and cadence.		5 to 7 min					40 min
Walk			10 min	March	40% to 80% of the HR_max_			
Stand up/sit down from chair + elastic bands	Sitting, with legs bent, forming a 90° angle between leg and thigh, feet flat on the floor at shoulder width, lift. Perform full trunk extension. The elastic must always be attached under the feet and held with the hands—Quadriceps’s Activation.			Max. concentric eccentric 3 s	6–9 OMNI-RES scale	15	3	
Low row + elastic bands	Seated; arms outstretched in order to grip the elastic. Pull your elbows toward your torso, keeping your back straight and your chest high—Back Activation							
Unilateral knee extension + shin guards	Seated; straight back; knees bent; feet resting on the ground. Carry out the extension of the same, until forming a parallel line between the lower limbs and the ground—Quadricep Activation							
Chest press + elastics	Sitting or standing; arms in pronation, flexed, and elbows in line with the shoulders. Carry out the extension of the same until the level of the chest. Look ahead. The elastic band can be on the dorsal area or on a post/pillar—Activation of the Big Pectoral.							
Single leg curl + shin guards	Standing. Flex your knees until you form a 90° angle between the thigh and the leg—Activation of the hamstrings							
High row + elastic	Standing; upper limbs extended at shoulder level and parallel to the ground; grab the elastic. Flex your upper limbs in order to pull the elastic towards your chest.							
Note: the rest interval between exercises or sets is 30 s.								
Flexibility: dynamic active method (stretching).								
Exercise	Duration (second)	Series	Recovery		Volume			
4 static stretches	30 s	1	Does not exist		2 min			
Note: stretching performed unilaterally, must last 15 s per limb (not in the case of this 1st micro cycle).								

## Data Availability

Not applicable.

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
