# Peer review of "Effects of Physical Exercise Program in Adults with Intellectual and Developmental Disabilities—A Study Protocol"

_jcm, 2022, doi:10.3390/jcm11247485_

Round 1
Reviewer 1 Report
Congratulations to the authors. The work is written in detail. (especially the method part). It is aimed to develop new exercise programs for adults with Intellectual and Developmental Disabilities. It has been evaluated as a study that contributes to the field. But some minor adjustments need to be made;
- In the last paragraph of the introduction, hypotheses should be explained.
-Line 80, The research design to be used in the study should be explained. Experimental design expression is insufficient.
-Line 88-93: How long after the last workout, especially the 2nd and 3rd measurements will be performed. It should be explained.
-In which time of the day will the three planned measurements be made? It should be explained.
-In Table 1 and Table 2, 40-80% of HHR, What does HHR mean? I think it is written wrongly.
-How will the control group be controlled for 24 weeks (because the time is long). It should be explained.
-What time of the day will the exercise programs be applied? It should be explained.
-Line 212: body composition
- Will the participants be informed about the criteria proposed by the ACSM before the body composition measurement?
-How will it be determined which exercise program is more effective on the parameters for statistical analysis? It should be explained.
-Why the correlation analysis will be done and between which parameters. Should it be disclosed? The statistical analysis section should be written in more detail and more clearly. Analyzes that are considered to be done should be reviewed.
-The discussion section should be written in more detail. It is written superficially. Possible mechanisms can be explained.
Author Response
Response to REVIEWER 1
Congratulations to the authors. The work is written in detail. (especially the method part). It is aimed to develop new exercise programs for adults with Intellectual and Developmental Disabilities. It has been evaluated as a study that contributes to the field. But some minor adjustments need to be made;
Response: Thank you very much for the thoughtful and insightful comments and appreciation that allowed us to greatly improve the quality of the manuscript. In the following, we highlight your concerns and we corrected the point by point manuscript accordingly.
In the last paragraph of the introduction, hypotheses should be explained.
Response: Thank you very much for your comment. The hypotheses under study have been described.
-Line 80, The research design to be used in the study should be explained. Experimental design expression is insufficient.
Response: An experimental, non-randomized study with 1:1 allocation was designed. This information is now included in the study design as required.
-Line 88-93: How long after the last workout, especially the 2nd and 3rd measurements will be performed. It should be explained.
Response: The measurements will be performed after 48 hours after the last workout in both groups. The study design included this information (lines 104-105).
-In which time of the day will the three planned measurements be made? It should be explained.
Response: The measurements will always be taken in the morning. We have added this information in lines 103-105.
-In Table 1 and Table 2, 40-80% of HHR, What does HHR mean? I think it is written wrongly.
Response: We thank the reviewer for his/her comment. Indeed, we agree with the reviewer. Therefore, the abbreviation has been corrected.
-How will the control group be controlled for 24 weeks (because the time is long). It should be explained.
Response: Participants in the control group will be encouraged to do their normal activities and the intervention group. Participation in the exercise groups (experimental group) will not be allowed. In addition to their normal activities performed in the institution, the exercise groups will engage in one of these two exercise programs, which will not be allowed for the control group (lines 93-98).
-What time of the day will the exercise programs be applied? It should be explained.
Response: The exercise programs will always be applied at the same time, namely in the morning. The existence of routines in this population is very important because it is structuring, reduces anxiety, and facilitates the acquisition of important concepts/knowledge/experience through repetition and knowledge of what is going to happen (line 13).
-Line 212: body composition
Response: Thank you, the word has been corrected.
- Will the participants be informed about the criteria proposed by the ACSM before the body composition measurement?
Response: Yes, it very important to conduct a comprehensive explanation of the study by the research leader and the host institution in order to allow participants/members/family members/guardians to be fully informed. This explanation also consisted of information about the data collection procedure, not only of the body composition but of all the tests performed. This explanation was also given at the evaluation moments (lines 133-138 and 212-214).
-How will it be determined which exercise program is more effective on the parameters for statistical analysis? It should be explained.
Response: The existence of significant differences between groups will be analyzed using the Kruskal-Wallis test (lines 400-401).
-Why the correlation analysis will be done and between which parameters. Should it be disclosed? The statistical analysis section should be written in more detail and more clearly. Analyzes that are considered to be done should be reviewed.
Response: Information has been added to this section, namely the justification for our choices.
-The discussion section should be written in more detail. It is written superficially. Possible mechanisms can be explained.
Response: We thank the reviewer for his/her comment. Thus, the discussion section has been revised
Reviewer 2 Report
This is an interesting article about a protocol of effects of physical exercise program in adults with intellectual and developmental disabilities.
Study design:
- Are the subjects in the gym and outdoor intervention groups going to perform the same exercises?
- A measurement could be added at 8-24 weeks after the last intervention to assess the long-term effects of this training.
Participants:
- In inclusion criterion 2, add an example of a medical contraindication.
- Inclusion criteria 1 and 4 are similar.
- Add in exlusion criteria contraindications to physical exercise.
- The minimum 15 subjects that have been calculated as the minimum sample size to carry out the study would be per group or among the 3 groups.
Protocols:
- Don't you think that since the protocols in the gym and outside the gym are so different, the type of exercise, the load, etc., could have an influence?
- Perhaps one way would be to perform before adjusting the load to each individual an aerobic test with a heart rate monitor and a strength test for each exercise in order to work with similar percentages for each participant.
- I think the difference in load between the groups training in and out of the gym is going to be a key factor in the results of this study, which is why in the comment above if aerobic and strength intensity could be calculated, it would give the study more consistency.
Statistical analysis
- You use non-parametric statistics because of the sample size, but you do not consider that even if it is a small sample the variable may follow a normal distribution and parametric tests should be used?
Outcomes
- On page 8, line 212, change the phrase "3.2. Body composition" to "3.2. Body composition".
Author Response
Response to REVIEWER 2
This is an interesting article about a protocol for the effects of physical exercise program in adults with intellectual and developmental disabilities.
Response: Thank you very much for the thoughtful and insightful comments and appreciation that allowed us to improve significantly the quality of the manuscript. In the following, we highlight your concerns and we corrected point by point the manuscript accordingly.
Study design:
Are the subjects in the gym and outdoor intervention groups going to perform the same exercises?
Response: We thank the reviewer for his/her comment. As described in table 1 and 2, the types of exercises will be different, although the movements and the working muscle groups are the same. I would clarify that the muscular groups used to perform the exercises are the same, however, they are executed using different equipment/materials in different environments (i.e., gym vs. outdoor context). Therefore, our intention is to compare the interventions.
The Measurement could be added at 8-24 weeks after the last intervention to assess the long-term effects of this training.
Response: We thank the reviewer for his/her comment. Indeed, the statement made by the reviewer is quite pertinent and a good suggestion. However, the aim of the present study is only related to the training process. Therefore, the detraining measure is not included in this protocol.
Participants:
In inclusion criterion 2, add an example of a medical contraindication.
Response: Thank you for your suggestion. By mistake, one of the inclusion criteria was duplicated in the exclusion criteria (medical contraindication/ contraindications to PE). We removed medical contraindications and added examples of contraindications to PE (line 121).
Inclusion criteria 1 and 4 are similar.
Response: The inclusion criteria have been reformulated.
Add in exclusion criteria contraindications to physical exercise.
Response: Thank you, it has been added.
The minimum 15 subjects that have been calculated as the minimum sample size to carry out the study would be per group or among the 3 groups.
Response: We thank the reviewer for his/her comment. Based on G*Power 3.1.9.2 (Faul et al., 2007) and the following input parameters: Effect size f (0.5 – medium effect size); α=0.05; Power (1-β err prob) = 0.95; the number of groups= 3; the number of measurements=3; corr among rep measures= 0.5; nonsphericity correction= 1. The required total sample size was 15. In addition, we included these parameters based on Denis's (2019) recommendations and on previous studies (e.g., Bartlo et al., 2011; Obrusnikova et al., 2021; St. John et al., 2020).
Protocols:
Don't you think that since the protocols in the gym and outside the gym are so different, the type of exercise, the load, etc., could have an influence?
Response: We agree with the reviewer. However, since financial limitations are considered to be one of the barriers to exercise for this population, our intention it to build an outdoor program (which does not require a subscription or monthly fee), as identical as possible to the one performed in a gym, with the intention of comparing it.
Perhaps one way would be to perform before adjusting the load to each individual an aerobic test with a heart rate monitor and a strength test for each exercise in order to work with similar percentages for each participant.
Response: The percentage of aerobic work in each exercise session was the same for all individuals in both programs (40-80% of HRmax). The indoor group (who will train with gym equipment), performed a maximal strength test in order to train according to the load percentage. This maximum strength test was not performed in the outdoor group since they did not have access to the gym and strength training will be performed with other types of equipment.
I think the difference in load between the groups training in and out of the gym is going to be a key factor in the results of this study, which is why in the comment above if aerobic and strength intensity could be calculated, it would give the study more consistency.
Response: We thank the reviewer for his/her comment. Indeed, we agree with reviewer's opinion. However, due to financial limitations, many institutions that provide support to this population do not provide physical exercise activities. Our intention was to create an exercise program with minimal economic expenditure so that the barrier presented above would be reduced. A priori, we think that indoor training (with greater rigor) may be more effective than outdoor training. However, studies have shown that outdoor training may be more effective in other groups (e.g., Brito, H. S., Carraça, E. V., Palmeira, A. L., Ferreira, J. P., Vleck, V., & Araújo, D. (2022). Benefits to Performance and Well-Being of Nature-Based Exercise: A Critical Systematic Review and Meta-Analysis. Environmental Science & Technology, 56(1), 62–77. https://doi.org/10.1021/acs.est.1c05151). The major goal of this project will be to evaluate the effectiveness of two interventions in different contexts. If they are effective, it would be possible to develop two different interventions. This can be crucial for institutions to replicate either one, according to their economic possibilities.
Statistical analysis
You use non-parametric statistics because of the sample size, but you do not consider that even if it is a small sample the variable may follow a normal distribution and parametric tests should be used?
Response: We thank the reviewer for his/her comment. However, and based on several authors (e.g., Hair et al., 2019; Fischer et al., 2011), to achieve a normal distribution it is necessary at least 30 subjects considering the rules of thumb of the central limit theorem. So, to be more conservative non-parametric tests will be used. By the way, this justification was added (lines 403-405).
Outcomes
On page 8, line 212, change the phrase "3.2. Body composition" to "3.2. Body composition".
Response: Thank you, the word has been corrected.
We would like to sincerely thank you for your advice and constructive comments.